# Bio-Mapping *Salmonella* and *Campylobacter* Loads in Three Commercial Broiler Processing Facilities in the United States to Identify Strategic Intervention Points

**DOI:** 10.3390/foods13020180

**Published:** 2024-01-05

**Authors:** Daniela R. Chavez-Velado, David A. Vargas, Marcos X. Sanchez-Plata

**Affiliations:** International Center for Food Industry Excellence, Department of Animal and Food Sciences, Texas Tech University, Lubbock, TX 79409, USA; daniela.r.chavez@ttu.edu (D.R.C.-V.); andres.vargas@ttu.edu (D.A.V.)

**Keywords:** poultry bio-mapping, *Salmonella* enumeration, *Campylobacter* enumeration

## Abstract

The poultry industry in the United States is one of the largest in the world. Poultry consumption has significantly increase since the COVID-19 pandemic and is predicted to increase over 16% between 2021 and 2030. Two of the most significant causes of hospitalizations and death in the United States are highly related to poultry consumption. The FSIS regulates poultry processing, enforcing microbial performance standards based on *Salmonella* and *Campylobacter* prevalence in poultry processing establishments. This prevalence approach by itself is not a good indicator of food safety. More studies have shown that it is important to evaluate quantification along with prevalence, but there is not much information about poultry mapping using quantification and prevalence. In this study, enumeration and prevalence of *Salmonella* and *Campylobacter* were evaluated throughout the process at three different plants in the United States. Important locations were selected in this study to evaluate the effect of differences interventions. Even though there were high differences between the prevalences in the processes, some of the counts were not significantly different, and they were effective in maintaining pathogens at safe levels. Some of the results showed that the intervention and/or process were not well controlled, and they were not effective in controlling pathogens. This study shows that every plant environment is different, and every plant should be encouraged to implement a bio-mapping study. Quantification of pathogens leads to appropriate risk assessment, where physical and chemical interventions can be aimed at specific processing points with higher pathogen concentrations using different concentrations of overall process improvement.

## 1. Introduction

In 2020, the United States produced over 20,000 metric tons of chicken. Currently, the poultry industry in the United States is the largest producer in the world after China, Brazil, and the European Union [1]. The total value of production in 2020 was United States Dollars (USD) 35.5 billion, with USD 21.6 billion being from broilers, USD 52 billion from turkey, USD 18.8 billion from chickens and USD 8.6 billion from eggs [2]. The United States is the second largest exporter of poultry behind Brazil. Poultry exports in the United States have had an average growth of 1.6 percent year-over-year since 2000. In 2020, exports were over 9 billion pounds and are expected to grow over the next 10 years, primarily due to broiler exports [3].

Poultry consumption during the COVID-19 pandemic in the U.S. increased by USD 1.3 billion in retail sales, according to research carried out by IRI, and 210 analytics show that during 2020 48% of the people surveyed said that they had increased the amount of chicken they consumed, and 39% responded that they had increased the ways they prepared chicken [4]. Global poultry consumption is predicted to increase by 16.3% between 2021 and 2030, and the meat trade between 2022 and 2030 is projected to be led by poultry [3].

Thirty-one major recognized pathogens cause an estimated 9.4 million illnesses, 55,961 hospitalization admissions, and 1351 deaths annually in the United States [5]. Norovirus is attributed to 58% of the illnesses, followed by nontyphoidal *Salmonella* spp. at 11%, 10% for *Clostridium perfringens*, and 9% for *Campylobacter*. Nontyphoidal *Salmonella* spp. is the main cause of hospitalization and mortality, causing 35% of hospitalizations and 28% of deaths [6]. More than 75% of the illnesses caused by *Salmonella* were assigned to seven food groups consisting of chicken, fruit, pork, seeded vegetable, nuts, turkey, and eggs. From these categories, 16.8% of the illnesses caused by *Salmonella* were attributed to chicken consumption. Illnesses caused by *Campylobacter* were more often attributed to chicken (64.7%) [7].

As a component of a comprehensive strategy to reduce the public impact of *Salmonella* infections caused by raw poultry products, the FSIS has regulated poultry slaughter, and processing establishments are required to develop a hazard analysis where they can determine the “food safety hazards that can occur before, during, and after entry into the establishment” along with pre-harvest interventions, sufficient sanitization measures during processing, and appropriate sanitary dressing practices at slaughter [8]. Based on the frequency (positive or negative) of *Salmonella* and *Campylobacter* in chicken processing facilities, the FSIS implements microbiological performance guidelines. Following the chilling step, one sample of a whole bird carcass and samples or parts are taken for this verification system once a week. The results are then inputted into a 52-week moving window database, which computes the performance of each individual plant and divides establishments into three categories [9]. Category 1 is defined as consistent process control and contains those establishments that have achieved 50% or less *Salmonella* or *Campylobacter* maximum allowable percent positive (presence in 9.8% of broiler carcasses, 25% of comminuted, 15.4% of chicken parts for *Salmonella*; and 1.9% of comminuted chicken and 7.7% of chicken parts for *Campylobacter*) during each of the last three months of the full 52-week moving windows. Establishments falling under category 2 are defined as variable process control and are those that have results greater that 50% of the maximum allowable percent positive during any completed 52-week moving window over the previous three months, even though they have met the *Salmonella* and *Campylobacter* maximum allowable percent positive for all completed 52-week moving windows. Establishments that have exceeded the maximum permitted percent positive for *Salmonella* or *Campylobacter* during any 52-week moving window during the previous three months are included in category 3, which denotes a highly variable process control [9].

Poultry carcass contamination with fecal material and pathogens like *Salmonella* is a hazard *reasonably likely to occur* (RLTO) according to the FSIS [10]. *Salmonella* can be controlled in poultry processing facilities through an HACCP plan or prevented in the processing environment through sanitation operating procedures (SSOPs) or other prerequisite programs [8]. Therefore, the focus of most chicken processors in the U.S. is in reducing the prevalence of *Salmonella* and *Campylobacter*, implementing sanitary dressing procedures, applying antimicrobial interventions to lower cross-contamination during processing and handling, and complying with regulatory performance standards using process controls.

Although the USDA-FSIS regulates zero fecal contamination and a carcass temperature of less than 40 °F when it exits the chiller, best practices for pathogen reduction for poultry processing has not been established. While antimicrobial type and amount used to control *Salmonella* and *Campylobacter* varies greatly among plants, scalder tanks, IOBW (inside–outside bird wash), a pre-chiller, and a primary immersion chiller are usually standard intervention points [11]. The relative low cost and low concentration required for efficacy of chlorine has historically made it a common antimicrobial utilized during different stages of the poultry processing line, including IOBW, pre-chiller, main chillers, and in post-chill applications [12]. However, studies have reported that the efficacy of chlorine for bacterial reduction is highly affected by the pH and organic load [13]. Currently, the most-used antimicrobial/processing aid in chillers and post-chill immersion is peracetic or peroxyacetic acid (PAA). This mixture of acetic acid and hydrogen peroxide is used for pathogen control during broiler processing and is an effective antimicrobial agent against bacteria and bacteria spores [14].

Process mapping studies, also known as bio-mapping studies, offer a standardized methodology to produce the evidence required to determine how best to use PAA and chlorine as well as many other interventions. It also focuses on the locations of interventions within the process, highlighting areas that can immediately be improved or that require process adjustments in order to maximize efficacy and improve the microbial performance and the overall effectiveness of the food safety plan. Establishments are heavily encouraged by the FSIS to employ bio-mapping studies to create their own pathogen and indicator microorganisms sampling programs [8].

In spite of the multiple approaches of poultry processors to accomplish USDA-FSIS performance standards, there is limited information about the quantification of *Salmonella* spp. and *Campylobacter* in comparing enumeration and prevalence percentage at different stages throughout the process. This study is one of the pioneers to integrate prevalence along with quantification of *Salmonella* and *Campylobacter* to create a baseline for three different plants to implement statistical process control parameters for each processing plant, and locations were developed to improve the food safety system.

## 2. Materials and Methods

### 2.1. Sample Collection

Three commercial processing facilities were sampled for this study using the USDA-FSIS performance standards sampling scheme [15]. These plants voluntarily requested a bio-mapping study due to the need to generate data in plants for risk assessment. Plant number one was sampled in spring 2022, plant two and three were sampled in fall 2022. For plant number one, nine locations throughout the processing line were sampled, including live receiving (LR), rehang (RH), post-evisceration (PE), pre-chill (PRE), post-chill (POST), pre-dip wings portions (PDW), packaging wings (W), pre-dip tenders (PDT), and packaging tenders (PT). At each location, 10 samples were taken per repetition; a total of 540 samples were taken during a six-day period. For plants number two and three, seven locations were sampled throughout the processing line, including live receiving (LR), rehang (RH), pre-chill (PRE), post-chill (POST), breast (BS), thighs (T), and wings (W). For plant number two, 2 samples were taken per day with two sampling days per week for 13 weeks, making a total of 364 samples. For plant number three, 2 samples were taken per day with two sampling days per week for 11.5 weeks, making a total of 322 samples.

Rinses were chilled and immediately shipped to the Texas Tech University Food Microbiology Laboratory of the International Center for Food Industry Excellence (ICFIE) for microbiological analysis.

### 2.2. Tempo Campylobacter Enumeration and Prevalence

Only plant number one was analyzed for *Campylobacter* since this was the only plant that requested *Campylobacter* data in their bio-mapping. For enumeration and prevalence using the TEMPO system [16], the ISO 16140/AFNOR method was used, the TEMPO cards were incubated for 44–48 h at 42 ± 1 °C under microaerophilic conditions using BD^®^ GasPak^®^ EZ container system sachets [17].

### 2.3. BAX^®^ Salmonella Enumeration and Prevalence

Upon arrival, samples were homogenized by hand and processed following the AOAC 081201 method for enumeration of *Salmonella* using the BAX^®^ System SalQuant™ [18], 30 mL of samples were combined with 30 mL of BAX MP with 40 mg of Novobiocin per liter [18] and placed in an incubator at 42 °C for 6 h for recovery. After enumeration, samples were placed in the incubator at 42 °C for an additional 18 h. Negative samples at enumeration were processed following the BAX^®^ System RT-*Salmonella* Assay for detection.

### 2.4. Limit of Quantification for Salmonella Results

As De Villena et. al. explained in their study, the limit of quantification (LOQ) for SalQuant is 1 CFU/mL; however, because counts can be extrapolated below LOQ, because they are derived from a regression equation supplied by the methodology, a new LOQ of 1% of the actual LOQ (0.01 CFU/mL or 0.6 log CFU/carcass) was established. A total of 50% of the new limit of quantification (0.3 log CFU/carcass) was reported for samples that showed <0.6 log CFU/carcass. For samples that could not be quantified but tested positive for prevalence, the same value was used. Samples with neither detectable nor quantifiable data were giving a 0 log CFU/carcass value [19].

### 2.5. Statistical Analysis

All data were analyzed using R (version 4.0.4) statistical analysis software to evaluate the reduction of microbial loads between each of the nine processing points for plant one and between each of the seven points for plant two and three. All counts were transformed to log CFU/mL for *Campylobacter* for plant number one; counts were reported as log CFU/carcass for *Salmonella* for all plants. A Kruskal–Wallis test (*p* < 0.01) was chosen as a nonparametric alternative of the ANOVA due to a lack of parametric assumptions. This was performed to compare counts at each location and a mean separation was performed through a Mann–Whiney *U* test or Wilcoxon rank sum test whenever the Kruskal–Wallis test was found to be significant.

## 3. Results

### 3.1. Salmonella Detection and Enumeration

#### 3.1.1. Salmonella Plant One

The average incoming load of *Salmonella* at the live receiving area was 2.39 log CFU/sample (Table 1). These counts were prior to any intervention or treatment at the processing plant and were obtained by sampling the most recent, warm, and healthy identified dead-on-arrival (DOA) birds. There was no significant difference between the live receiving and the rehanger point for this plant (*p* > 0.01), but there was a difference between live receiving and post-evisceration, with a mean reduction of 0.8 log CFU/sample. The lowest counts obtained in the processing area in this plant were at the post-chill sampling point, with an average of 0.25 log CFU/sample. The parts sampling portion for this plant—pre-dip wings portions, wings, pre-dip tenders, and packaging tenders—were statistically different except for the pre-dip wings portions and the pre-dip tenders. The lowest count at parts was found for the packaging tenders, with a mean of 0.11 log CFU/sample.

For prevalence (Figure 1), the highest presence of *Salmonella* was found at the rehanger location and at the pre-dip wings portions, with an 88.33% presence of *Salmonella* for both. The lowest presence was found at the post-chill location, with a 35.00% prevalence of *Salmonella*.

#### 3.1.2. *Salmonella* Plant Two

For plant number two, the average incoming load of *Salmonella* was 2.83 log CFU/sample (Table 2) at the live receiving stage, where no treatment or intervention was made. The counts for live receiving were obtained by sampling the most recent dead-on-arrival (DOA), warm, and identified healthy birds. There was a significant difference between the live receiving and the rehanger point for this plant (*p* < 0.01); the mean difference was 2.15 log CFU/sample. All the sampling points during processing were significantly different except between rehang and pre-chill. The mean reduction between pre-chill and post-chill was 0.64 log CFU/carcass. For the parts (breast, thighs, and wings), there was no statistical difference, with *p* = 0.75 being the lowest *p*-value among them.

The presence of *Salmonella* was higher at the processing locations and lower at the part portions sampled. The higher prevalence of *Salmonella* was found at the live receiving (98.08%) (Figure 2), followed by the rehanger (61.54%). The lower presence of *Salmonella* was found for breast (5.77%), the post-chill location (7.69%), followed by thighs and wings (9.62%).

#### 3.1.3. *Salmonella* Plant Three

The incoming average load of *Salmonella* for plant number three was 2.78 log CFU/sample (Table 3), measured at the live receiving area. Significant differences were found among processing locations (*p* < 0.01), but no differences were found between after-chill and parts. The mean reduction between live receiving and the rehanger was 2.02 log CFU/sample. The lowest count was found at post-chill, with an average of 0.01 log CFU/sample, followed by thighs and wing parts with an average count of 0.03 log CFU/sample.

For the presence of *Salmonella* at this plant, the highest was found at the live receiving location, where 95.65% (Figure 3) of the samples were detected as positives, followed by the rehanger (73.91%). The lowest prevalence of *Salmonella* was found in the post-chill location (4.38%).

### 3.2. Campylobacter

*Campylobacter* enumeration was conducted only in plant number one; for this plant, the results indicate that there is a very low or non-existing presence of *Campylobacter*. The only location with the presence of *Campylobacter* was live receiving, with a mean of 0.20 log CFU/mL (Table 4). These counts were prior to any intervention. The other locations when *campylobacter* was suitable for counts were the rehanger (0.03 log CFU/mL) and post-evisceration (0.01 log CFU/mL).

## 4. Discussion

One of the current approaches to food safety in the poultry industry consists of pathogen reduction and process control (Hazard Analysis and Critical Control) to reduce the risk of consumer exposure to foodborne pathogens, such as *Salmonella* and *Campylobacter* [20]. The success of this is based on compliance with performance standards based on pathogen prevalence [21]. Because of this, many studies conducted on *Salmonella* and *Campylobacter* in poultry have focused the microbial reduction based only on prevalence (%) or the number of positives without evaluating what is the concentration per sample or carcass. Along with the study conducted by De Villena et al. (2022), these two studies are the pioneers in evaluating the quantification of *Salmonella* and *Campylobacter* along with their prevalence in poultry processing facilities.

Although prevalence testing is the most frequent approach to food safety in the poultry industry, *Salmonella* or *Campylobacter* prevalence alone is not a good indicator of poultry food safety. According to previous studies [21,22,23], pathogen prevalence is only one of many risk factors that determine food safety. Based on the results, all processing plants in this study indicate a reduction in *Salmonella* prevalence and quantification along the processing line.

In the poultry processing industry, processors follow different approaches to control and reduce pathogen prevalences along the processing, which include the application of antimicrobials at different steps in the processing line along with physical interventions [21]. According to other studies, a one logarithmic pathogen reduction from location to location in a poultry processing facility is necessary to consider an intervention effective [22,23,24,25,26,27].

### 4.1. Salmonella Prevalence and Quantification Plant One

Plant number one was the one with the most consistently high prevalence of *Salmonella* of all the plants included in this study, and where the prevalence at the rehanger location and pre-dip wings portions (88.33%) were higher than the live receiving prevalence (86.67%). The rehanger samples are those taken after the scalding processing; this is where carcasses are placed in hot water to facilitate feather removal and is the first location during processing where positive *Salmonella* carcasses can spread *Salmonella* to negative carcasses through cross-contamination [28]. Moreover, during the picking process cross-contamination can occur because of contact with contaminated rubber picking fingers and contaminated reuse water [29,30].

During the grow-out period, the intestinal tract of chickens may harbor *Salmonella*, and damage to the gut during processing can cross-contaminate the carcasses and processing equipment [31,32]. These results indicate an opportunity for equipment cleaning and disinfection improvement for this plant. Samples from the live receiving location included feathers, feet, heads, dirt, and fecal contamination from the fields, which are often highly contaminated with *Salmonella* [33,34]. This can decrease when scalding is well controlled. For this plant, there was not a significant difference (*p* = 0.55042) between the live receiving location and the rehanger location; the mean reduction was 0.54 log CFU/sample. As described in previous studies, high levels of *Salmonella* found on incoming birds can overwhelm establishment intervention, and these levels of contamination can be carried forward to the next steps of the process [35].

Although there was no statistical difference (*p* = 0.17661) between pre- and post-chill locations, the average counts of *Salmonella* (0.40 log CFU/sample and 0.25 log CFU/sample), as well as wings (0.29 log CFU/sample) and packaging tenders (0.11 log CFU/sample) according to risk assessment of *Salmonella* in broiler chicken, these levels have a very low possibility of causing illness if handled and cooked appropriately to avoid cross-contamination [36,37].

### 4.2. Salmonella Prevalence and Quantification Plant Two

Reductions of *Salmonella* prevalence observed in other studies are attributed to sequential washes and antimicrobial interventions applied at the evisceration step and chilling tanks [38,39]. For plant number two, the significant log reduction of *Salmonella* from live receiving to the rehanger (2.83 log CFU/sample to 0.68 log CFU/sample) provides validation that the scalding and picking processes are important steps for bacterial reduction during the process if properly managed. This agrees with other studies that found significant reductions even without any pH adjustment treatments in the scalder tanks [19,40]. Although the mean difference in count from live receiving and the rehanger was significant, with a mean difference of 2.15 log CFU/sample, the difference in prevalence between these two points was 36.54% (98.08% to 61.54%), confirming that prevalence alone is not a good indicator of food safety since reduction went from 2.83 log CFU/sample to a safe level of 0.68 log CFU/sample, but the presence of *Salmonella* is still high (61.54%) [23].

Samples taken post-chiller show a significant log reduction from 0.69 log CFU/sample at pre-chill to 0.05 log CFU/sample at post-chill. The pattern of prevalence was very similar, changing from 59.62% to 7.69%. Immersion in the chilling tank is the major antimicrobial intervention used during poultry processing for pathogen reduction [41]. According to other studies, parts typically have a higher *Salmonella* prevalence than the chilled carcass, which is the reason why they are treated again with antimicrobials before packaging [38,41]. For this plant, there was no significant difference between parts; the lowest prevalence was found for breast at 5.77%, with a mean of 0.10 log CFU/sample, and the highest prevalence at parts was for things and wings, with 9.62% and a mean of 0.17 log CFU/sample and 0.03 log CFU/sample, respectively, showing that even though they have the same 9.62% presence, quantification shows that they both are at safe levels, but wings had the lowest counts.

### 4.3. Salmonella Prevalence and Quantification Plant Three

Plant number three started with a relatively high prevalence (95.65%) at the live receiving location. There was a statistical difference (*p* < 0.01) in *Salmonella* counts between live receiving and the rehanger location (2.78 log CFU/sample and 0.76 log CFU/sample). This again supports the importance of the scalding process and impact of the removal of feathers, head, feet, and dirt from the field when is properly managed. There was also a significant difference between the rehanger and pre-chill (0.76 log CFU/sample and 0.30 log CFU/sample), showing that the evisceration process is well monitored in this plant and the interventions implemented between the rehanger and the pre-chill locations are effective at reducing *Salmonella* concentrations. The GI tract is a major source of pathogen contamination during evisceration, and this is an important point inspected by the FSIS to look for visible fecal contamination. The post-chill location had the lowest prevalence of *Salmonella* (4.38%). This correlates with the mean counts at this point in the process (0.01 log CFU/sample). This plant uses a pre-pre-chiller, with a concentration of 180 ppm of PAA, a pre-chiller (100 ppm PAA), and a main chiller (100 ppm PAA), which has been shown to be efficient in reducing Salmonella concentration to safe levels.

### 4.4. Campylobacter Quantification Plant One

For plant number one, there was no available data to see levels of *Campylobacter* due to its low or non-existing presence in this plant. Counts were found at live receiving (0.20 log CFU/mL), the rehanger (0.03 log CFU/mL), and post-evisceration (0.01 log CFU/mL). These results do not relate to other studies, where up to 7.9 log CFU/mL of *Campylobacter* was found in poultry samples [33].

## 5. Conclusions

Pathogen quantification can result in appropriate risk assessment, where physical and chemical interventions can be targeted to specific stages with higher pathogen concentrations using different concentrations of overall process improvement. This study provides evidence for the application of chemical schemes at different stages of processing, tailoring interventions for higher-risk areas. The development of bio-mapping baselines will result in statistical process control analysis to support food safety management decision-making. In this study, there is evidence that each plant environment is different, and as a result, each plant should be encouraged to perform a baseline study to optimize their interventions’ efficacies, reduce costs, and have a more robust food safety plant overall, thereby preventing foodborne illness.

The results for plant one point to a need for better equipment cleaning and disinfection. Re-evaluating and modifying the efficacy of the interventions at this facility is necessary to lower the amount of Salmonella present in the final product. According to the results found in this study, the interventions, and controls at plants two and three are effective in reducing the prevalence and quantification levels of Salmonella to acceptable levels.

## Figures and Tables

**Figure 1 foods-13-00180-f001:**
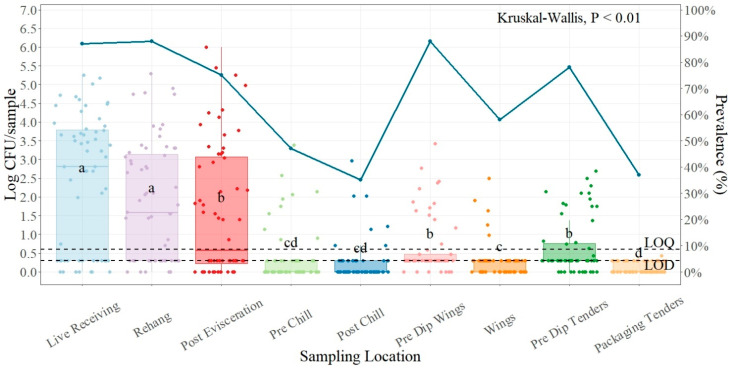
*Salmonella* counts (log CFU/sample) and prevalence (shown as solid blue line) comparison between sampling points during processing and parts. In every box plot, the median is represented by the horizontal line, and the upper and lower vertical lines, respectively, stand for 1.5 times the upper and lower interquartile ranges. The data points are represented by the dots. ^a–d^ for each location, boxes with distinct letters show statistically different between sampling points (*p*-value < 0.01). Limit of quantification for SalQuant (LOQ) is 1% of the real LOQ (0.6 log CFU/carcass), and limit of detection (LOD) is 50% of the new LOQ (0.3 log CFU/sample), both shown as dotted lines.

**Figure 2 foods-13-00180-f002:**
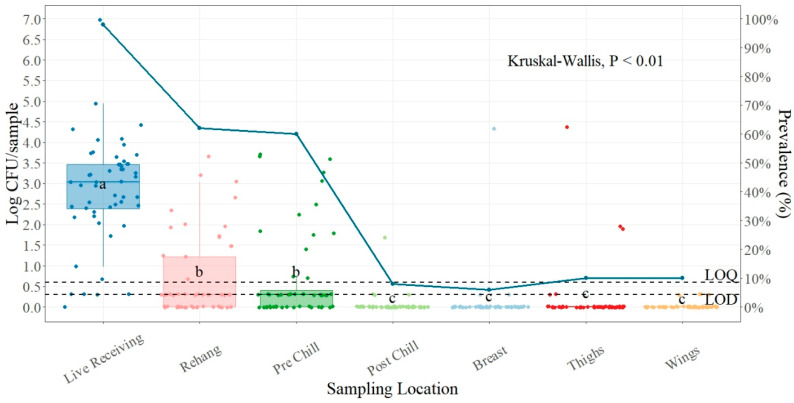
*Salmonella* counts (log CFU/sample) and prevalence (shown as solid blue line) comparison between sampling points during processing and parts. In every box plot, the median is represented by the horizontal line, and the upper and lower vertical lines, respectively, stand for 1.5 times the upper and lower interquartile ranges. The data points are represented by the dots. ^a–c^ for each location, boxes with distinct letters show statistically different between sampling points (*p*-value < 0.01). Limit of quantification for SalQuant (LOQ) is 1% of the real LOQ (0.6 log CFU/carcass), and limit of detection (LOD) is 50% of the new LOQ (0.3 log CFU/sample), both shown as dotted lines.

**Figure 3 foods-13-00180-f003:**
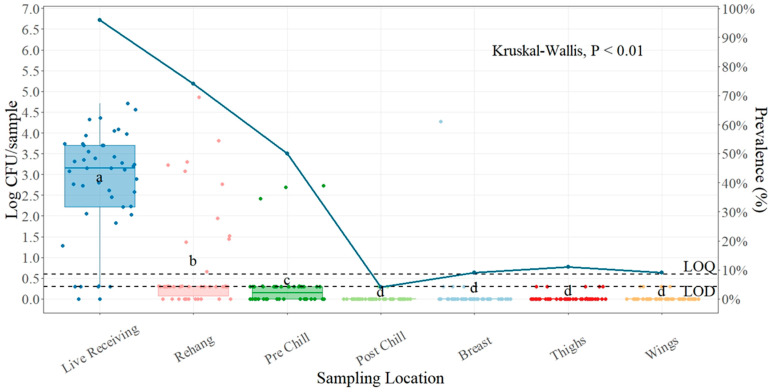
*Salmonella* counts (log CFU/sample) and prevalence (shown as solid blue line) comparison between sampling points during processing and parts. In every box plot, the median is represented by the horizontal line, and the upper and lower vertical lines, respectively, stand for 1.5 times the upper and lower interquartile ranges. The data points are represented by the dots. ^a–d^ for each location, boxes with distinct letters show statistically different between sampling points (*p*-value < 0.01). Limit of quantification for SalQuant (LOQ) is 1% of the real LOQ (0.6 log CFU/carcass), and limit of detection (LOD) is 50% of the new LOQ (0.3 log CFU/sample), both shown as dotted lines.

**Table 1 foods-13-00180-t001:** *Salmonella* counts (log CFU/sample) and prevalence (%) at nine locations during processing stages and part portions of chicken rinses for plant number one.

	*Salmonella* (*n* = 540)	
Location	Counts (Log CFU/Sample ± SE ^1^)	Prevalence %
Live Receiving	2.39 ± 0.23 ^a^	52/60 (86.67)
Rehanger	1.85 ± 0.20 ^a^	53/60 (88.33)
Post Evisceration	1.59 ± 0.23 ^b^	44/60 (75.00)
Pre-Chill	0.40 ± 0.09 ^cd^	28/60 (46.67)
Post-Chill	0.25 ± 0.07 ^cd^	21/60 (35.00)
Pre-Dip Wings	0.63 ± 0.10 ^b^	53/60 (88.33)
Wings	0.29 ± 0.06 ^c^	35/60 (58.33)
Pre-Dip Tenders	0.63 ± 0.10 ^b^	47/60 (78.33)
Packaging Tenders	0.11 ± 0.02 ^d^	22/60 (36.67)

^1^ Standard error of the mean; ^a–d^ For each location, different letters between locations are significantly different (*p*-value < 0.01).

**Table 2 foods-13-00180-t002:** *Salmonella* counts (log CFU/sample) and prevalence (%) at seven locations during processing stages and part portions of chicken rinses for plant number two.

	*Salmonella* (*n* = 364)	
Location	Counts (Log CFU/Sample ± SE ^1^)	Prevalence %
Live Receiving	2.83 ± 0.18 ^a^	51/52 (98.08)
Rehanger	0.68 ± 0.14 ^b^	32/52 (61.54)
Pre-Chill	0.69 ± 0.15 ^b^	31/52 (59.62)
Post-Chill	0.05 ± 0.03 ^c^	4/52 (7.69)
Breast	0.10 ± 0.08 ^c^	3/52 (5.77)
Thighs	0.17 ± 0.10 ^c^	5/52 (9.62)
Wings	0.03 ± 0.01 ^c^	5/52 (9.62)

^1^ Standard error of the mean; ^a–c^ for each location, different letters between location are significantly different (*p*-value < 0.01).

**Table 3 foods-13-00180-t003:** *Salmonella* counts (log CFU/sample) and prevalence (%) at each of the seven locations during processing stages and part portions of chicken rinses for plant number three.

	*Salmonella* (*n* = 322)	
Location	Counts (Log CFU/Sample ± SE ^1^)	Prevalence %
Live Receiving	2.78 ± 0.19 ^a^	44/46 (95.65)
Rehanger	0.76 ± 0.17 ^b^	34/46 (73.91)
Pre-Chill	0.30 ± 0.09 ^c^	23/46 (50.00)
Post-Chill	0.01 ± 0.01 ^d^	2/46 (4.38)
Breast	0.11 ± 0.09 ^d^	4/46 (8.70)
Thighs	0.03 ± 0.01 ^d^	5/46 (10.67)
Wings	0.03 ± 0.01 ^d^	4/46 (8.70)

^1^ Standard error of the mean; ^a–d^ for each location, different letters between location are significantly different (*p*-value < 0.01).

**Table 4 foods-13-00180-t004:** *Campylobacter* counts (log CFU/mL) at each of the nine locations during processing stages and part portions of chicken rinses for plant number one.

	*Campylobacter* (*n* = 540)
Location	Counts (Log CFU/mL ± SE)
Live Receiving	0.20 ± 0.61
Rehanger	0.03 ± 0.14
Post-Evisceration	0.01 ± 0.07
Pre-Chill	0.00 ± 0.00
Post-Chill	0.00 ± 0.00
Pre-Dip Wings Portions	0.00 ± 0.00
Wings	0.00 ± 0.00
Pre-Dip Tenders	0.00 ± 0.00
Packaging Tenders	0.00 ± 0.00

## Data Availability

Data is contained within the article.

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
