# Peer review of "Bio-Mapping Salmonella and Campylobacter Loads in Three Commercial Broiler Processing Facilities in the United States to Identify Strategic Intervention Points"

_foods, 2024, doi:10.3390/foods13020180_

Round 1

Reviewer 1 Report

Comments and Suggestions for Authors

The manuscript by Chavez-Velado et al. reports about the Salmonella and Campylobacter bacterial quantification in commercial broiler processing facilities. The study appears interesting, focusing the attention on the different stages throughout the process of poultry slaughtering.

Some minor concerns are reported below. The authors should respond to these comments.

In material and methods section, at point 2.1 and 2.2 (lines 118 and 135), authors should clarify which where the differences between these three plant facilities, why the choose these three plants and in which season the sampling was performed.

Author Response

Thank you so much for your feed back. I added the clarifications in the manuscript as the following:

On line 118 (120): These plants voluntarily requested bio-mapping study due to the need to generate data in plants for risk assessment. Plant number one was sampling spring 2022, plant two and three were sampling on fall 2022.

On line 132 (139): Only plant number one was analyzed for Campylobacter since this was the only plant that requested Campylobacter data in their bio-mapping.

Reviewer 2 Report

Comments and Suggestions for Authors

The objective of this manuscript is to demonstrate the efficacy of implementing a standardized approach that would facilitate the determination of management procedures within the industry, specifically in a slaughterhouse, to minimize risk. This study is interesting because it provides data that would allow for a better evaluation of the risk of the two pathogens studied, particularly for Salmonella. The study of Campylobacter has not been done correctly.

Some aspects should be improved or answered:

1) In the analysis of Campylobacter, only enumeration was carried out, which means that the prevalence cannot be evaluated.

2) Why was p < 0.01 set as a significance value? It is usually p < 0.05.

3) Is it possible to combine the results of the different plants into a single table, evaluating the statistical differences between them? This approach may determine the causes and offer recommendations for maintaining the current control measures or proposing new strategies to enhance food safety oversight across various processing steps.

4) The figures provide the same results as the tables. Figures are more visual, but the information is complex to see in them.

5) Further clarification is needed on the concept of the 1.5-timer interquartile range, as its significance is not apparent. If relevant, it should be explained in the text. Perhaps the figures would be unnecessary.

6) The manuscript would benefit from a more thorough discussion of the results. The feeling is that the results and the discussion are similar. Based on the results obtained, it would be interesting to apply quantitative microbial risk assessment methods to evaluate the effectiveness of control measures

Author Response

Thank you so much for your comments, your feedback is really appreciated and constructive, I proceed to respond of each comment and made the corresponding changes in the manuscript, please read the responses below:

  • In the analysis of Campylobacter, only enumeration was carried out, which means that the prevalence cannot be evaluated. For the Campylobacter study, only plant number one was analyzed for Campylobacter since this was the only plant that requested Campylobacter data in their bio-mapping. The method used for Enumeration was the TEMPO system, this method results are present as Log CFU/mL. The prevalence was obtained by calculating how many of the total of samples analyzed were countable (positives) and how many were below the limit of quantification of the methodology (LOQ = 1CFU/mL) (negatives).
  • Why was p < 0.01 set as a significance value? It is usually p < 0.05. The level of significance for an experiment is decided by the authors previously to perform the statistical analysis. During conversations before starting the experiment authors decided to use an alpha value equal to 0.01 to be stricter at detecting differences between groups, especially during multiple comparison. By using 0.01 as a level of significance there is more confidence that the differences between groups are not found by random chance.
  • Is it possible to combine the results of the different plants into a single table, evaluating the statistical differences between them? This approach may determine the causes and offer recommendations for maintaining the current control measures or proposing new strategies to enhance food safety oversight across various processing steps. Yes, it is possible to combine the results of each plant, but we cannot compare the results between them, since each plant is a different environment with different interventions, different concentrations of chemicals, different flocks, line speeds and, they were sampling on different seasons (Plant 1: spring, Plant 2 and 3: fall).
  • The figures provide the same results as the tables. Figures are more visual, but the information is complex to see in them. We appreciate the comment of the reviewer, however the information was added as figures and tables with the purpose of having a full description of the results (Table) as well as a visual perspective of the results to identify patterns or clear locations for continuous improvements.
  • Further clarification is needed on the concept of the 1.5-timer interquartile range, as its significance is not apparent. If relevant, it should be explained in the text. Perhaps the figures would be unnecessary. Boxplots are a type of visualization where multiple levels of a specific variable can be visualized as well as their respective distribution. A boxplot shows measurements of central tendency (median) as well as measurements of variability (quartiles). Moreover, some statistics can be also visualized as the IQR (interquartile range) that allows you to classify the data from a distribution into possible outliers, reason why authors decided to present the results in this format to have a better picture of the distribution of the data.
  • The manuscript would benefit from a more thorough discussion of the results. The feeling is that the results and the discussion are similar. Based on the results obtained, it would be interesting to apply quantitative microbial risk assessment methods to evaluate the effectiveness of control measures. We completely agree with the reviewer’s statement, and we decided to include more information into the discussion section. Moreover, since the objective of the study is to create a baseline and provide information about Salmonella and Campylobacter prevalence and quantification in different steps in broilers processing facilities, quantitative microbial risk assessment is out of the scope of the present study. Nevertheless, authors are considering doing a data mining of the current results where risk analysis will be considered.

Round 2

Reviewer 2 Report

Comments and Suggestions for Authors

The authors’ responses are satisfactory and the discussion is improved. They emphasize the importance of establishing the prevalence and quantifying the possible pathogenic microorganisms for a proper risk assessment, which is the basis for implementing and monitoring control procedures to minimize the risk.

The high prevalence of plant 1 indicates a problem. The low concentration in the final product is still significant enough to cause cross-contamination. If the cold chain fails, the concentration will increase, possibly leading to a foodborne illness. Therefore, I disagree with the authors’ statement in line 489.

The control measures in the three plants may differ. What is the authors’ final conclusion about the effectiveness of the control measures that mainly affect the prevalence of Salmonella?

Author Response

Thank you so much for you comments, please see my responses below:

I has not able to find line 489 (its a reference in my document). however, I added to plant number one statements that it is true if the product with that concentration/prevalence is handled and cooked carefully since that is what my references for that stated.  

For the next question: 

The control measures in the three plants may differ. What is the authors’ final conclusion about the effectiveness of the control measures that mainly affect the prevalence of Salmonella?

This study provides evidence that every plant is a different environment, as a result, each plant should be encouraged to perform a baseline study to optimize their interventions efficacies. 

The results for plant one point to a need for better equipment cleaning and disinfection. Reevaluating and modifying the efficacy of the interventions in this facility is necessary to lower the amount of Salmonella present in the final product. According to the results found in this study, plants two and three interventions and controls are effective in reducing the prevalence and quantification levels of Salmonella to acceptable levels.